**Distribution and rates of nitrogen fixation in the western tropical South Pacific Ocean constrained by nitrogen isotope budgets**

Angela N. Knapp[1], Kelly M. McCabe[1], Olivier Grosso[2], Nathalie Leblond[3], Thierry Moutin[2], and Sophie Bonnet[4]

[1]Earth, Ocean, and Atmospheric Science Dept., Florida State University, 117 N. Woodward AVE., Tallahassee, FL 32306, USA.
[2]Aix-Marseille Université, CNRS, Université de Toulon, IRD, OSU Pythéas, Mediterranean Institute of Oceanography (MIO), UM 110, 13288, Marseille, France
[3]Observatoire Océanologique de Villefranche, Laboratoire d'Océanographie de Villefranche, UMR 7093, Villefranche-sur-mer, France
[4]Aix-Marseille Université, CNRS, Université de Toulon, IRD, OSU Pythéas, Mediterranean Institute of Oceanography (MIO), UM 110, 98848 Noumea, New Caledonia

*Correspondence to*: Angela N. Knapp (anknapp@fsu.edu)

**Abstract.** Constraining the rates and spatial distribution of di-nitrogen ($N_2$) fixation fluxes to the ocean informs our understanding of the environmental sensitivities of $N_2$ fixation as well as the timescale over which the fluxes of nitrogen (N) to and from the ocean may respond to each other. Here we quantify rates of $N_2$ fixation as well as its contribution to export production along a zonal transect in the Western Tropical South Pacific (WTSP) Ocean using N isotope ("$\delta^{15}N$") budgets. Comparing measurements of water column nitrate+nitrite $\delta^{15}N$ with the $\delta^{15}N$ of sinking particulate N at a western, central, and eastern station, these $\delta^{15}N$ budgets indicate high, modest, and low rates of $N_2$ fixation at the respective stations. The results also imply that $N_2$ fixation supports exceptionally high, i.e., $\geq 50\%$, of export production at the western and central stations, which are also proximal to the largest iron sources. These geochemically-based rates of $N_2$ fixation are equal to or greater than those previously reported in the tropical North Atlantic, indicating that the WTSP Ocean has the capacity to support globally significant rates of $N_2$ fixation, which may compensate for N removal in the oxygen deficient zones of the eastern tropical Pacific.

**Key Points:**

- Higher $N_2$ fixation rates are observed at the western vs. eastern end of a zonal transect in the western tropical South Pacific

- Nitrogen isotope budgets indicate that $\geq$50% of export production was supported by $N_2$ fixation at stations with significant dissolved iron

- $N_2$ fixation in the western tropical southwest Pacific may compensate for N loss in the eastern tropical Pacific

# 1 Introduction

The primary source of nitrogen (N) to the ocean is the biologically-mediated reduction of di-nitrogen ($N_2$) gas to ammonia, which is then assimilated into the biomass of the organisms carrying out this process, known as diazotrophs (Gruber, 2004). While the distribution and rates of this process in the ocean play a central role in regulating the fertility and community structure of marine ecosystems, these first-order properties of marine $N_2$ fixation remain poorly constrained. Historically, the highest rates of $N_2$ fixation in the global ocean have been associated with the tropical North Atlantic (Mahaffey et al., 2005; Sohm et al., 2011). The high $^{15}N_2$ incubation-based $N_2$ fixation rates observed in the tropical Atlantic (Luo et al., 2012) are consistent with both the preference of diazotrophs for warm waters (Breitbarth et al., 2007; Stal, 2009) as well as the high atmospheric dust flux to the region (Mahowald et al., 2009; Prospero, 1996) that helps fulfil the high iron requirement of the enzyme, nitrogenase, carrying out $N_2$ fixation (Berman-Frank et al., 2001; Kustka et al., 2003). Additionally, the elevated ratio of nitrate ($NO_3^-$) to phosphate ($PO_4^{3-}$) concentrations (Gruber and Sarmiento, 1997) and low $\delta^{15}N$-$NO_3^-$ (Knapp et al., 2008) in the upper thermocline of the North Atlantic are attributed to high regional $N_2$ fixation rates, and have supported the hypothesis that iron availability plays a key role in regulating the spatial distribution of $N_2$ fixation in the ocean (Moore et al., 2009; Moore and Doney, 2007) ("$\delta^{15}N$", where $\delta^{15}N = \{[(^{15}N/^{14}N)_{sample}/(^{15}N/^{14}N)_{reference}] - 1\}*1000$, with atmospheric $N_2$ as the reference).

While the highest inputs of N to the ocean have traditionally been associated with the North Atlantic, it has also been argued that this association results from the significant sampling bias in favor of the tropical Atlantic (Sohm et al., 2011), with large regions of the South Pacific and Indian Ocean under-sampled with respect to direct $N_2$ fixation rate measurements (Luo et al., 2012). More recently, the Eastern Tropical South Pacific (ETSP) has seen increased sampling due to nutrient distribution-based modelling predictions that the highest global $N_2$ fixation rates would be found in surface waters above and adjacent to oxygen deficient zones (ODZs), where significant phosphorus (P) would be available to support $N_2$ fixation (Deutsch et al., 2007). However, field campaigns have found exceedingly low rates of $N_2$ fixation in the ETSP gyre (Knapp et al., 2016a; Raimbault and Garcia, 2008) (Moutin et al.,

2008), which have been attributed to limited iron availability (Dekaezemacker et al., 2013). Consequently, existing measurements indicate that the dominant sinks for N in the ocean, benthic and water column denitrification and anaerobic ammonium oxidation, focused in the ODZs of the eastern tropical Pacific and Arabian Sea (Gruber and Galloway, 2008), are spatially segregated from the

dominant $N_2$ fixation inputs in the tropical Atlantic. This spatial decoupling of N inputs and outputs necessarily corresponds to a temporal decoupling, requiring the time scale of ocean circulation for $N_2$ fixation to respond to changes in rates of denitrification, and vice versa. In spite of the apparent spatial decoupling in the modern ocean, paleoceanographic evidence indicates that N fluxes to and from the ocean have been closely balanced over $\geq 20$ kya, requiring feedbacks in the N cycle to operate on time

scales shorter than ocean circulation, and thus implying a tighter spatial coupling of N sources and sinks (Brandes and Devol, 2002; Deutsch et al., 2004). While N loss in the ocean is constrained to suboxic sediments and water column ODZs, similar constraints on the location of the largest $N_2$ fixation fluxes to the ocean are lacking, and thus the degree to which marine N sources and sinks have been coupled through time remains uncertain.

While prior modelling analyses emphasized the importance of iron *or* phosphorus in supporting $N_2$ fixation, the most recent modelling studies reflect the importance of elevated surface temperatures, adequate iron, *and* the potential for low surface ocean $NO_3^-:PO_4^{3-}$ concentration ratios to support a unique ecological niche for diazotrophs (Dutkiewicz et al., 2012; Monteiro et al., 2011; Weber and

Deutsch, 2014). Attention has consequently shifted to the relatively undersampled Western Tropical South Pacific (WTSP) Ocean, where atmospheric dust fluxes to warm surface waters are higher than in the central and eastern tropical South Pacific (Mahowald et al., 2009), and where surface ocean $NO_3^-$ and $PO_4^{3-}$ concentrations and ratios are relatively advantageous for diazotrophs (Moutin et al., 2005; Van Den Broeck et al., 2004). While seasonally some regions nearer to islands experience $PO_4^{3-}$

drawdown to lower levels (e.g., (Van Den Broeck et al., 2004), Moutin, this issue), in parts of the WTSP gyre surface ocean $NO_3^-$ concentrations are $\leq 0.1$ µM and $PO_4^{3-}$ concentrations are ~0.05 to 0.2 µM (Garcia et al., 2014), with corresponding positive P* values (where P* = $[PO_4^{3-}] - [NO_3^-]/16$) (Deutsch et al., 2007). Additionally, early remote sensing work detected significant and persistent

blooms of *Trichodesmium* spp. in the WTSP (Dupouy et al., 2000), consistent with more recent direct observations of elevated *Trichodesmium* spp. abundance and $N_2$ fixation rates observed near Melanesian islands (i.e., New Caledonia, Vanuatu, and Fiji) (Moisander et al., 2010; Shiozaki et al., 2014; Stenegren et al., 2018; Yoshikawa et al., 2015) and in the Solomon Sea (Bonnet et al., 2009; 2015: Berthelot et al., 2017). These high *Trichodesmium* spp. abundances and $N_2$ fixation rates have been attributed to sea surface temperatures >25 °C and continuous nutrient inputs of terrigenous and volcanic origin (Labatut et al., 2014; Radic et al., 2011). Prior molecular work has also shown higher rates of $N_2$ fixation in the WTSP at locations where surface ocean dissolved iron (DFe) concentrations were higher and where *Trichodesmium* spp. were less stressed for iron (Chappell et al., 2012). Together, these observations and modelling-based predictions highlight the potential for significant $N_2$ fixation rates in regions of the WTSP where diazotrophs can meet their iron and phosphorus requirements.

Here we use geochemical tools to quantify rates of $N_2$ fixation along a zonal transect in the WTSP where surface waters are ≥25 °C, have favourable macronutrient concentrations and ratios, and where DFe concentrations are an order of magnitude higher than in the South Pacific Gyre, and are mainly attributable to shallow hydrothermal input (Guieu et al., under review). We then compare these geochemical estimates of $N_2$ fixation rates with other metrics of $N_2$ fixation evaluated on this cruise, as well as with the global distribution of marine $N_2$ fixation rates.

## 2 Methods

### 2.1 Sample collection

Sampling for the Oligotrophic to UlTra-oligotrophic PACific Experiment ("OUTPACE") cruise was conducted on the R/V L'Atalante, which left Noumea, New Caledonia on 18 February 2015 and arrived in Papeete, Tahiti, on 2 April 2015. This cruise followed a roughly zonal transect along 18 to 19 °S between 159 °E and 160 °W. Details of the cruise and experimental design are described comprehensively in (Moutin et al., 2017), but briefly, sediment traps were deployed at three "Long Duration" (LD) stations A, B, and C (Table 1) (Fig. 1a). Water column samples were collected from

Niskin bottles deployed on a CTD-rosette at both LD as well as "short duration" (SD) stations (Fig. 1a), and water was stored at -20 ºC in HDPE bottles for analysis on land.

## 2.2 $NO_3^-+NO_2^-$ concentration and $\delta^{15}N$ measurements

The concentrations of $NO_3^-+NO_2^-$ in water column samples collected on the OUTPACE cruise were measured by colorimetric methods (Aminot and Kerouel, 2007). The $\delta^{15}N$ of $NO_3^-+NO_2^-$ in samples collected on the OUTPACE cruise was measured using the denitrifier method (Casciotti et al., 2002; Sigman et al., 2001), with modifications (McIlvin and Casciotti, 2011) (Fig. 1b). Typical standard deviation of the $NO_3^-+NO_2^-$ $\delta^{15}N$ analyses was $\leq$0.2‰, with error bars for individual analyses shown in Fig 1c.

## 2.3 Sinking particulate N flux and $\delta^{15}N$ measurements

Surface-tethered floating particle-interceptor traps (PPS5) were deployed on the OUTPACE cruise at 150, 330 and 520 m for ~5 days at Stations LD A and LD B, and at 150 and 330 m at LD C (Moutin et al., 2017). The mass flux ("$PN_{sink}$ flux") and $\delta^{15}N$ of the $PN_{sink}$ flux was determined by combustion-GC interfaced to an isotope ratio mass spectrometer at the Mediterranean Institution of Oceanography with a lower detection limit of 2.2 µg N and precision of $\pm$ 0.3‰ for 80 µg samples, with a precision of $\pm$ 1.0‰ for 10 to 20 µg samples typical of what was collected in the sediment traps at the LD stations.

## 2.4 $\delta^{15}N$ budget calculations

Here we compare the $\delta^{15}N$ of the two dominant sources of "new" N to surface waters, subsurface $NO_3^-$ and $N_2$ fixation, with the $\delta^{15}N$ of the sinking particulate N ($PN_{sink}$) flux to estimate the relative importance of both $NO_3^-$ and $N_2$ fixation as a source of new N to surface waters. This approach relies on subsurface $NO_3^-$ and $N_2$ fixation having distinct isotopic compositions. $N_2$ fixation introduces new N to the ocean with a $\delta^{15}N$ of ~ -1‰ (Carpenter et al., 1997; Hoering and Ford, 1960; Minagawa and Wada, 1986). In contrast, in the Pacific, $NO_3^-$ mixed up from the subsurface is impacted by water column denitrification and can have a $NO_3^-$ $\delta^{15}N$ >20‰ (e.g., (Brandes et al., 1998; Casciotti et al., 2013; Rafter and Sigman, 2016)), although as upper thermocline waters move westward in the Pacific, the very high $NO_3^-$ $\delta^{15}N$ signal is diluted and typical values are between 5 and 10‰ (Lehmann et al., 2018; Rafter et al., 2013). The relative importance of each source for supporting export production can be determined

using the two end-member mixing model described in Eqn. 1 ("$\delta^{15}N$ budget") where the fractional importance of $N_2$ fixation for supporting export production (x) is defined as:

$$PN_{sink} \; \delta^{15}N = x(-1‰) + (1 - x)(NO_3^-+NO_2^- \; \delta^{15}N) \qquad \text{Eq. 1}$$

Rearranging and solving for x yields:

$$x = (NO_3^-+NO_2^- \; \delta^{15}N – PN_{sink} \; \delta^{15}N)/(1 + NO_3^-+NO_2^- \; \delta^{15}N) \qquad \text{Eq. 2}$$

Multiplying the fraction of export production supported by $N_2$ fixation (x) by the $PN_{sink}$ mass flux provides a time-integrated $N_2$ fixation rate that can be compared with $^{15}N_2$ incubation-based $N_2$ fixation rate measurements (Knapp et al., 2016a). Here it is hypothesized that both rates of $N_2$ fixation and its importance for fuelling export production will be higher at stations in the western vs. central and eastern regions of the WTSP because of their closer proximity to iron sources (Guieu et al., under review).

## 3 Results

### 3.1 $NO_3^-+NO_2^-$ concentration and $\delta^{15}N$, and $PN_{sink} \; \delta^{15}N$

Samples collected in the upper 70 m at the LD stations had $\leq 0.1$ µM $NO_3^-+NO_2^-$ (Caffin et al., 2017) and increased with depth, consistent with prior regional observations (Garcia et al., 2014) (Fig. 1c). All nutrient concentration data are available at: http://www.obs-vlfr.fr/proof/php/outpace/outpace.php. Water column profiles of thermocline $NO_3^-+NO_2^- \; \delta^{15}N$ are available at: https://www.bco-dmo.org/dataset/733237/data and show similar trends at the LD stations, with 650 m $NO_3^-+NO_2^- \; \delta^{15}N$ ~7‰, increasing to ~8.5‰ at 400 m (Fig. 1b, 1c) (Knapp et al., 2018), which fall within the range of previous regional measurements (Yoshikawa et al., 2015). The elevation of thermocline $NO_3^-+NO_2^- \; \delta^{15}N$ relative to the mean ocean $NO_3^-+NO_2^- \; \delta^{15}N$ of 5‰ is attributed to denitrification and/or anammox occurring in the ODZs of the ETSP, where thermocline $NO_3^- \; \delta^{15}N$ can exceed 20‰ (e.g., (Altabet et al., 2012; Casciotti et al., 2013)). The average, mass-weighted $\delta^{15}N$ of the $PN_{sink}$ flux collected in the 150 m

trap increased from the western to eastern stations, from 0.6 ± 1.0‰ at LD A, to 3.1 ± 1.0‰ at LD B, and to 7.7 ± 1.0‰ at LD C (Table 1) (Fig. 1c).

## 3.2 Results of the $\delta^{15}N$ budget: $N_2$ fixation rates and their contribution to export production

Estimates of $N_2$ fixation rates and their contribution to export production determined using $\delta^{15}N$ budgets include the quantitatively dominant fluxes of N into and out of the surface ocean. Here, the dominant fluxes of N into the surface ocean include subsurface $NO_3^-$ and newly fixed N introduced from diazotrophs, and the dominant loss term is represented by the $PN_{sink}$ flux (Eq. 1). In the event that total dissolved N (TDN) concentrations vary in space/time, they may be included as well, however, surface ocean TDN concentrations from the OUTPACE cruise show little to no zonal gradient, and were typically between 5 and 7 µM in the upper 100 m (Moutin et al., This issue), and so are not included in $\delta^{15}N$ budget calculations. Additionally, the importance of N in atmospheric deposition has recently received significant attention, especially in the northwest Pacific (e.g., (Kim et al., 2014)), raising the possibility that atmospheric N deposition might also be an important source of N in the WTSP. However, the atmospheric N deposition flux measured on the OUTPACE cruise, 0.2 µmol N m$^{-2}$ d$^{-1}$ (Caffin et al., 2017), is several orders of magnitude lower than the mass flux captured in the 150 m sediment traps, 30 to 300 µmol N m$^{-2}$ d$^{-1}$ (Table 1), indicating that atmospheric N deposition is an insignificant source of new N to regional surface waters, and so is neglected in our $\delta^{15}N$ budget calculations.

While gradients with depth in subsurface $NO_3^-+NO_2^-$ $\delta^{15}N$ at the OUTPACE LD stations are modest compared to those in the ETSP, due to the relatively low sampling resolution in the upper thermocline where $NO_3^-$ is likely sourced, we calculate $\delta^{15}N$ budgets using a range of $NO_3^-+NO_2^-$ $\delta^{15}N$ end-member values, which are represented by the shaded regions in Fig. 1c. At each LD station, the $NO_3^-+NO_2^-$ $\delta^{15}N$ lower bound is represented by the 650 m sample and the upper bound is represented by the 400 m sample. Samples collected shallower than this (i.e., ≤200 m) are either within this range or show elevation in $NO_3^-+NO_2^-$ $\delta^{15}N$ as the $NO_3^-+NO_2^-$ concentration decreases, and reflects the effect of $NO_3^-$ assimilation, as is commonly observed below the euphotic zone in other oligotrophic regions (Knapp et al., 2016a; Knapp et al., 2008), and thus do not represent the $\delta^{15}N$ of the source $NO_3^-$. Using the $PN_{sink}$

$\delta^{15}N$ ($\pm$ 1‰ 1 S.D.) and the range in subsurface $NO_3^-+NO_2^-$ $\delta^{15}N$ end-member values in Eq. 2 corresponds to 80 to 83 $\pm$13%, 50 to 56 $\pm$12%, and 0 to 8 $\pm$11% of export production supported by $N_2$ fixation at stations LD A, LD B, and LD C, respectively (Table 1). Multiplying the fractional importance of $N_2$ fixation by the $PN_{sink}$ mass flux yields a range of estimated $N_2$ fixation rates of 219 to 290, 11 to 20, and 0 to 9 $\mu$mol N m$^{-2}$ d$^{-1}$ at stations LD A, LD B, and LD C, respectively (Table 1), where the range includes uncertainty in both the $PN_{sink}$ $\delta^{15}N$ measurement as well as the $NO_3^-+NO_2^-$ $\delta^{15}N$ end-member.

## 4 Discussion

### 4.1 Comparison of $\delta^{15}N$ budget results with other $N_2$ fixation metrics from the OUTPACE cruise

The $N_2$ fixation rates derived from the $\delta^{15}N$ budgets described above are lower than those measured by *in situ* $^{15}N_2$ incubations at the same OUTPACE stations, with depth-integrated average $N_2$ fixation rates of 593 $\pm$ 51, 706 $\pm$ 302, and 59 $\pm$ 16 $\mu$mol N m$^{-2}$ d$^{-1}$ at LD A, LD B, and LD C, respectively (Caffin et al., 2017). Previous work has also found lower $\delta^{15}N$ budget-derived $N_2$ fixation rates relative to $^{15}N_2$ incubation-based $N_2$ fixation rates (Knapp et al., 2016a). To the extent that sediment traps under collect the export flux, the two different metrics of $N_2$ fixation may be reconciled by multiplying "x" from Eq. 2, the fractional importance of $N_2$ fixation for export production, by other metrics of new or export production such as $O_2$/Ar ratios, $^{234}Th$ deficits, or $^{14}C$ uptake rates (Knapp et al., 2016a). This explanation may reconcile the $\delta^{15}N$ budget and $^{15}N_2$ incubation-based $N_2$ fixation rate estimates at LD A, which differ by a factor of ~2.5, and potentially the rates at LD C as well, which, while they differ by a factor $\geq$6, both correspond to relatively low $N_2$ fixation rates. However, the $\delta^{15}N$ budget and $^{15}N_2$ incubation-based $N_2$ fixation rates observed at LD B, 11 to 20 and 706 $\mu$mol N m$^{-2}$ d$^{-1}$, respectively, are more difficult to reconcile based on sediment trap under-collection alone, and may be partially attributable to variability encountered while sampling at the end of a phytoplankton bloom as well as the fate of newly fixed N at that station (Caffin et al., 2018; de Verneil et al., 2017). We note that the zonal trend in increasing $PN_{sink}$ $\delta^{15}N$ to the east is similar to a zonal gradient in suspended particulate N ($PN_{susp}$) $\delta^{15}N$ (Bonnet et al., This Issue), suggesting that the $\delta^{15}N$ of the $PN_{sink}$ observed at LD B is consistent with other regional geochemical data. Additionally, the $^{15}N_2$ incubation-based $N_2$ fixation

rate at LD B has relatively large error bars, resulting from observations of decreasing *in situ* $N_2$ fixation rates over the course of several daily observations at LD B (Caffin et al., 2017), which may also contribute to the offset between the $^{15}N_2$ incubation and $\delta^{15}N$ budget-based $N_2$ fixation rate estimates. Further, the $PN_{sink}$ flux collected in the 150 m trap at LD B, 0.030 mmol N $m^{-2}$ $d^{-1}$, was somewhat lower

than the $PN_{sink}$ flux collected in the 330 and 520 m traps at the same station, 0.034 and 0.036 mmol N $m^{-2}$ $d^{-1}$, respectively, which is unexpected given the more typical mass flux attenuation with depth observed at LD A and LD C, as well as elsewhere in the ocean (Martin et al., 1987). This unusual trend in mass flux with depth suggests either non-steady state sinking flux conditions (Caffin et al., 2018) and/or a problem with the sediment trap sample collection at LD B. Regardless, using the $^{14}C$-uptake

based estimate of net community production at LD B, 1.91 mmol N $m^{-2}$ $d^{-1}$, instead of the $PN_{sink}$ mass flux to multiply "x" from Eq. 2 by yields an $N_2$ fixation rate of 2300 µmol N $m^{-2}$ $d^{-1}$. These significant disparities in productivity metrics and resulting $N_2$ fixation rates at LD B suggests the potential for temporal decoupling of production and export and/or the underestimation of the export flux by the sediment trap, and indicate that $N_2$ fixation rates are probably higher than those resulting from $\delta^{15}N$

budget calculations based on the mass flux to the 150 m trap at LD B. Regardless, we take the zonal trend in $PN_{sink}$ $\delta^{15}N$ to indicate a decreasing contribution from $N_2$ fixation to export from the west to the east to be robust as it is consistent with both the $PN_{susp}$ $\delta^{15}N$ measurements as well as the broad trends in $^{15}N_2$ incubation-based $N_2$ fixation rate estimates that decrease from the west to east.

Comparing the absolute magnitude of the $\delta^{15}N$ budget-based $N_2$ fixation rates with previous measurements, we find that the 219 to 290 µmol N $m^{-2}$ $d^{-1}$ rate estimated for LD A represents a significant $N_2$ fixation rate relative to prior global measurements (Luo et al., 2012), in particular if it should be revised upwards to account for the under-collection of the export flux by the sediment trap. In contrast, the estimated rate range at LD B, 11 to 21 µmol N $m^{-2}$ $d^{-1}$, is quite low, as is the range of 0 to 9

µmol N $m^{-2}$ $d^{-1}$ at LD C, and both of these rates are broadly similar to the rates previously measured in the ETSP (Knapp et al., 2016a; Moutin et al., 2008; Raimbault and Garcia, 2008). Similarly, the $\delta^{15}N$-budget based estimate of the contribution of $N_2$ fixation to export production at LD C is low and similar to previous $\delta^{15}N$-budget measurements in the North Pacific (Casciotti et al., 2008) and North Atlantic

(Altabet, 1988; Knapp et al., 2005). However, the fractional contribution of $N_2$ fixation to export production at LD A, 80 to 83%, is higher than all previous $\delta^{15}N$ budget results. The contribution of $N_2$ fixation to export production at LD B, 50 to 57%, is also notably high. While the previous $\delta^{15}N$ budgets of (Karl et al., 1997) and (Dore et al., 2002) found evidence for ~50% of export production supported by $N_2$ fixation near Hawaii, newer methods capable of measuring the $NO_3^-+NO_2^-$ $\delta^{15}N$ at the lower $NO_3^-$ $+NO_2^-$ concentrations found in the upper thermocline that represent a more realistic estimate of the end-member $NO_3^-$ source suggest that $N_2$ fixation may support closer to 25% of export during the summer in the North Pacific gyre (Bottjer et al., 2017; Casciotti et al., 2008). Consequently, the findings of 50 to 57% and 80 to 83% of export production being supported by $N_2$ fixation at stations LD B and LD A, respectively, indicates that $N_2$ fixation plays a significant role supporting carbon fixation and export production in this region of the WTSP, consistent with the high e-ratios (up to 9.7) reported by (Caffin et al., 2017). Direct export of diazotrophs has been reported by (Caffin et al., 2017), but most export is likely indirect, i.e., after the transfer of diazotroph-derived N to non-diazotrophic plankton, that is subsequently exported (Caffin et al., 2018), as has been observed elsewhere in the WTSP (Bonnet et al., 2016; Knapp et al., 2016b).

**4.2 Environmental sensitivities of $N_2$ fixation and the basin-scale coupling of N sources and sinks**

The zonal gradient in both $N_2$ fixation rates as well as their contribution to export production in the OUTPACE study supports emerging hypotheses regarding the controls on the distribution of marine $N_2$ fixation fluxes in the global ocean. Specifically, the low rates of $N_2$ fixation documented in this study at LD C and in the ETSP (Knapp et al., 2016a) indicate that low $NO_3^-:PO_4^{3-}$ concentration ratios in the absence of adequate iron (Blain et al., 2008; Fitzsimmons et al., 2014) are insufficient to support significant fluxes of new N to the ocean. Instead, the results presented here are consistent with recent modelling work that has included *both* the high iron requirements of diazotrophs as well as the potential for low $NO_3^-:PO_4^{3-}$ concentration ratios to support elevated diazotroph abundance and $N_2$ fixation inputs to the ocean (Dutkiewicz et al., 2012; Monteiro et al., 2011; Weber and Deutsch, 2014). Indeed, these new modelling efforts have identified the WTSP as a unique region where $PO_4^{3-}$ concentrations are relatively high, $NO_3^-$ concentrations are low, and atmospheric dust fluxes provide a moderate source of

iron to warm surface waters, conditions seemingly favourable for significant $N_2$ fixation fluxes. While regions within the WTSP nearer to islands experience significant $PO_4^{3-}$ drawdown, with seasonal $PO_4^{3-}$ turnover times comparable to those observed in the Sargasso Sea (Van Den Broeck et al., 2004; Van Mooy et al., 2009), these modelling predictions are supported by recent reports of high regional $^{15}N_2$ incubation-based $N_2$ fixation rates (Bonnet et al., 2017).

However, prior to the OUTPACE cruise, our knowledge of DFe concentrations and their sources in the WTSP was limited, especially in the western and central sectors. During OUTPACE, Guieu et al. (Under review) reported high DFe concentrations in the western sector of the WTSP (from 160 °E to 165 °W, average 1.7 nM within the photic layer), i.e., significantly ($p<0.05$) higher than those reported in the eastern sector (165 °W to 160 °W, average 0.3 nM within the photic layer). The high DFe concentrations measured in the west were previously undocumented, and reveal several maxima (>50 nM), suggesting significant iron inputs to this region. (Guieu et al., Under review) found that atmospheric deposition in this region was too low to explain the observed DFe concentrations in the water column, and that the iron in the euphotic layer may instead derive from shallow (~500 m) hydrothermal sources associated with the Tonga-Kermadec subduction zone.

Recent studies performed in the western end of the WTSP in the Solomon, Bismarck (Berthelot et al., 2017; Bonnet et al., 2009; Bonnet et al., 2015) and Arafura (Messer et al., 2015; Montoya et al., 2004) Seas also reveal extremely high $N_2$ fixation rates (>600 µmol N m$^{-2}$ d$^{-1}$), indicating that high $N_2$ fixation rates have been found over a significant region of the WTSP, extending west to east from Australia to Tonga and north to south from the equator to 25 to 30 °S, or ~13 x $10^6$ km$^2$ (i.e. ~20% of the South Pacific Ocean area). These significant N inputs may offset the N loss occurring in the ODZs of the eastern tropical Pacific. The ability for marine N inputs and outputs to compensate for each other within the same ocean basin corresponds to a spatial and thus temporal coupling on the scale of years to decades, consistent with the paleoceanographic record (Brandes and Devol, 2002; Deutsch et al., 2004; Weber and Deutsch, 2014), and represents an intermediate view of the distribution of global marine $N_2$ fixation fluxes consistent with that proposed by (Weber and Deutsch, 2014) where iron availability

controls local $N_2$ fixation rates but phosphorus availability regulates basin-scale $N_2$ fixation rates (Moutin et al., 2008 and this issue).

**5 Conclusions**

The goal of this study was to address the question: do regions other than the tropical Atlantic contribute significantly to global $N_2$ fixation fluxes? While our results should be taken as a "snapshot" view that cannot necessarily be scaled up to annual fluxes, at stations proximal to iron sources, geochemically-derived $N_2$ fixation rates of 219 to 290 μmol N m$^{-2}$ d$^{-1}$ were observed, and could potentially represent a lower bound of $N_2$ fixation rates due to the potential under-collection of the $PN_{sink}$ flux by sediment traps. Moreover, at stations LD A and LD B, separated by ~27° longitude, $N_2$ fixation was found to support >50% of export production, a finding that has not been replicated elsewhere with sensitive $NO_3^-$ +$NO_2^-$ $\delta^{15}N$ methods to our knowledge. Together with similar findings from $^{15}N_2$ uptake experiments, these results suggests that $N_2$ fixation can support a significant fraction of export production over a large region of the WTSP. At the eastern station most distant from iron sources, both rates and the contribution of $N_2$ fixation to export production were low, ~0 to 9 μmol N m$^{-2}$ d$^{-1}$ and 0 to 8%, respectively, similar to previous measurements in the ETSP where diazotrophs may also be challenged by iron availability (Dekaezemacker et al., 2013; Knapp et al., 2016a; Moutin et al., 2008). Significant $N_2$ fixation fluxes in the WTSP may provide a means of balancing N loss occurring in the ODZs of the eastern tropical Pacific, and thus may help reconcile the paleoceanographic record requiring N inputs and losses to balance each other on time scales shorter than ocean circulation (Dutkiewicz et al., 2012; Monteiro et al., 2011; Weber and Deutsch, 2014).

**Acknowledgments, Samples, and Data**

This is a contribution of the OUTPACE (Oligotrophy to Ultra-oligoTrophy PACific Experiment) project (https://outpace.mio.univ-amu.fr/) funded by the French research national agency (ANR-14-CE01-0007-01), the LEFE-CyBER program (CNRS-INSU), the GOPS program (IRD) and the CNES. Funding for the contribution of A.N. Knapp was provided by the U.S. National Science Foundation OCE #1537314. The OUTPACE cruise (http://dx.doi.org/10.17600/15000900) was managed by the MIO (OSU Institut Pytheas, AMU) from Marseilles (France). The authors thank the crew of the R/V

L'Atalante for outstanding shipboard operation. G. Rougier and M. Picheral are warmly thanked for their efficient help in CTD rosette management and data processing, as is Catherine Schmechtig for the LEFE CYBER database management and C. Miranda for analytical assistance. All data and metadata are available at the following web address: http://www.obs-vlfr.fr/proof/php/outpace/outpace.php. The

5   data supporting the conclusions of this paper may be obtained at the BCO-DMO database https://www.bco-dmo.org/dataset/733237. The authors declare no other affiliations or real or perceived financial conflicts of interest with respect to the results of this paper.

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

**Table 1.** Location, subsurface $NO_3^-+NO_2^-$ $\delta^{15}N$, $PN_{sink}$ $\delta^{15}N$, and $N_2$ fixation rate and contribution to export at the OUTPACE long duration stations.

**Figure 1.** Map of the OUTPACE cruise with "long duration" (LD) stations A, B, and C noted (a), water column $NO_3^-+NO_2^-$ $\delta^{15}N$ measurements from the OUTPACE cruise (b), and CTD fluorescence (green line), $NO_3^-+NO_2^-$ concentration (filled circles), $NO_3^-+NO_2^-$ $\delta^{15}N$ (open circles), and $PN_{sink}$ $\delta^{15}N$ (filled inverted triangles) from OUTPACE stations LD A (c), LD B (d), and LD C (e). Error bars represent 1

standard deviation, and are smaller than the symbol size for $NO_3^-+NO_2^-$ concentration and most $NO_3^-$ $+NO_2^-$ $\delta^{15}N$ analyses. The range of $NO_3^-+NO_2^-$ $\delta^{15}N$ end-member values used for $\delta^{15}N$ budget calculations are represented by the shaded regions. The $N_2$ fixation end-member $\delta^{15}N$ value, -1‰, is represented by the arrows on the upper x-axis.

Table 1. Location, subsurface $NO_3^-+NO_2^-$ $\delta^{15}N$, $PN_{sink}$ $\delta^{15}N$, and $N_2$ fixation rate and contribution to export at the OUTPACE long duration stations.

| Station | Latitude (°N) | Longitude (°E) | Average $PN_{sink}$ flux (µmol N m⁻² d⁻¹) | 150 m Trap $PN_{sink}$ $\delta^{15}N$* (‰) | Subsurface $NO_3^-+NO_2^-$ $\delta^{15}N$ (‰) | % Export $N_2$ fixation | $N_2$ fixation rate (µmol N m⁻² d⁻¹) |
|---|---|---|---|---|---|---|---|
| LDA | -19.22 | 163.59 | 303 | 0.6 ± 1 | 7.0 to 8.4 | 80 to 83 ± 13% | 219 to 290 |
| LDB | -18.18 | -170.74 | 30 | 3.1 ± 1 | 7.2 to 8.3 | 50 to 56 ± 12% | 11 to 20 |
| LDC | -18.5 | -165.79 | 47 | 7.7 ± 1 | 7.0 to 8.4 | 0 to 8 ± 11% | 0 to 9 |

*Flux-weighted mean $PN_{sink}$ $\delta^{15}N$

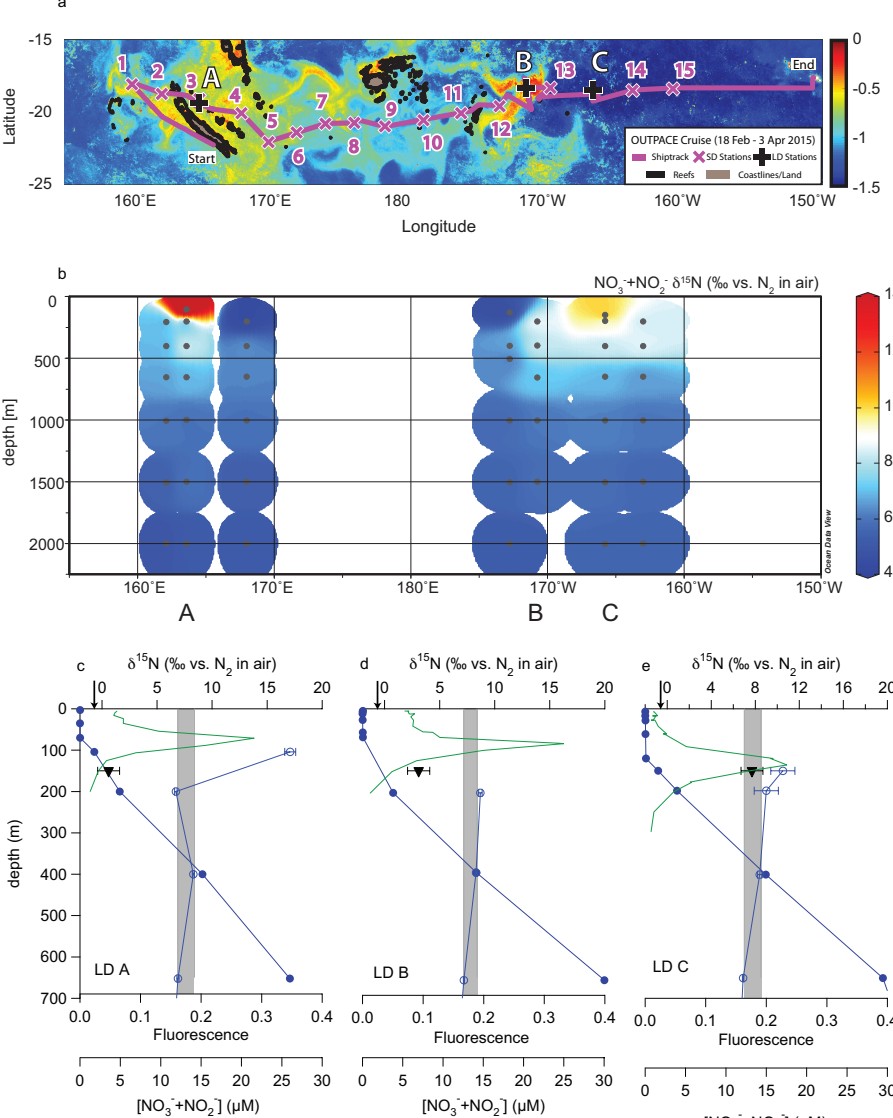