# Peer review of "Distribution and rates of nitrogen fixation in the western tropical South Pacific Ocean constrained by nitrogen isotope budgets"

_Biogeosciences, 2017_

## Referee Comment (RC1) · Anonymous Referee #1 · 6 Feb 2018

The manuscript by Angela Knapp and colleagues describes rates of nitrogen fixation via the analysis of nitrogen isotopes. This is a topic that Angela knows very well and has published multiple times on previous occasions. It is therefore not surprising that the manuscript is well-written with a good knowledge of the relevant literature. I recommend the manuscript for publication with just a few minor comments

Specific comments 1. This work seems to suggest there is an offset of N isotope analysis from to in situ 15N incubations which might be resolved or understood with time, rather than just analytical differences leading to variable results. Is this the case or am I just optimistic? 2. Some of the geochemical description on Page 5 Lines 13-28

and Page 6 Lines 1-12 could go into the Methods section which could use the extra length 3. Some broader context for the sediment traps might be helpful - where was the 150 m depth horizon with regards to the 1% light level or the DCM or the nutricline? 4. I realize OUTPACE has multiple publications, but a map showing station locations would be useful to quickly orientate the reader without having to look it up.

---

## Referee Comment (RC2) · Anonymous Referee #2 · 6 Feb 2018

The manuscript by Knapp and colleagues estimates the input of N via N2 fixation using a biogeochemical approach in the western south Pacific Ocean. The manuscript is well-written, and explains the conclusions well, addressing the relevant references. However, this reviewer considers that a few minor changes need to be addressed before publication.

Specific comments.

I do agree with Reviewer 1 in two points. First, though it is clear that this manuscript is related to many other publications coming from OUTPACE probably showing a map of station, having the map here will be useful for readers beginning from this work, instead

of having to look for the geographical context on their own.

Second, it will be useful to extend a bit the context of the sediment traps, maybe adding an additional line in Fig 1 showing either the mixed layer depth or the 1

The units of Average PNsink flux in Table 1 and in the Results section are not the same. The text is in $\mu$mol N, and the table in mmol N. I recommend units in agreement in both parts.

Some parts of section 3.2 could be moved to the Methods section, while others seem to fit better in the Discussion, as a first subsection 4.1. My suggestion is that P7 L25 to P8 L14 and P9 L4 to P11 L7 move to the Discussion, while P8 L15 to L23 up to "of the source NO3" move to the Methods. This way the Results subsection is reduced to the description of the results themselves (P8 L23 to P9 L3).

P3 L26. Just curiosity, but is there a reason for using the term Oxygen Deficient Zones instead of the mosth widely used Oxygen Minimun Zones (OMZs)?

P5-P6. The description of the geochemical tools could be moved to the Methods section. And it could be more intuitive to begin the name of the variables by $\delta$15N-xx. It is a bit confusing reading NO3+NO2 $\delta$15N, for instance.

P7 L16. What do the authors mean with "thermocline NO3+NO2"? Do they refer to subsurface NO3+NO2 as in section 3.2, or NOx produced in the thermocline?

---

## Author Comment (AC1) · 15 Apr 2018

Response to Reviewers We thank the reviewers for their constructive feedback. Based on their suggestions, the following changes have been made:

We have included a figure illustrating the location of the field work (Fig. 1a), as well as included additional data to provide more context for the results. The additional data include a section plot showing the NO3-+NO2- $\delta$15N at additional stations across the transect (Fig. 1b), as well as the fluorescence trace on the figure of the $\delta$15N budgets to indicate the range in depth and magnitude of productivity within the euphotic zone (Fig. 1c).

[Figure]

Additional changes in response to specific comments are described below in italics.

Review #1: The manuscript by Angela Knapp and colleagues describes rates of nitrogen fixation via the analysis of nitrogen isotopes. This is a topic that Angela knows very well and has published multiple times on previous occasions. It is therefore not surprising that the manuscript is well-written with a good knowledge of the relevant literature. I recommend the manuscript for publication with just a few minor comments.

Specific comments 1. This work seems to suggest there is an offset of N isotope analysis from to in situ 15N incubations which might be resolved or understood with time, rather than just analytical differences leading to variable results. Is this the case or am I just optimistic?

The reviewer is correct that recent work shows an offset between N2 fixation rates estimated by 15N2 bottle uptake experiments and $\delta$15N budgets. A discussion of potential causes for this offset can be found in Knapp et al., 2016, PNAS. Some factors specific to the OUTPACE study that potentially contributed to the lower N2 fixation rates estimated by $\delta$15N budgets than those estimated by 15N2 bottle uptake experiments are addressed in the text briefly, and include potential decoupling during a large Trichodesmium spp. bloom at LD B, as well as the fate of that newly fixed N, which is further discussed in de Verniel et al., 2018, as well as Caffin et al., 2018, both this issue.

2. Some of the geochemical description on Page 5 Lines 13-28 and Page 6 Lines 1-12 could go into the Methods section which could use the extra length

The text has been modified as suggested, and this text is now section 2.4 in the Methods.

3. Some broader context for the sediment traps might be helpful - where was the 150 m depth horizon with regards to the 1% light level or the DCM or the nutricline?

We appreciate the Reviewer's interest in a more complete representation of the biogeochemistry of the sediment trap stations. We have included the fluorescence trace in Fig. 1c, which can be compared with the NO3-+NO2- concentration and $\delta$15N profiles, and have included more information in Figure 1 as described above.

4. I realize OUTPACE has multiple publications, but a map showing station locations would be useful to quickly orientate the reader without having to look it up.

We appreciate the feedback and have now included a map of the station locations (Fig. 1a).

---

## Author Comment (AC2) · 15 Apr 2018

Response to Reviewers We thank the reviewers for their constructive feedback. Based on their suggestions, the following changes have been made:

We have included a figure illustrating the location of the field work (Fig. 1a), as well as included additional data to provide more context for the results. The additional data include a section plot showing the NO3-+NO2- $\delta$15N at additional stations across the transect (Fig. 1b), as well as the fluorescence trace on the figure of the $\delta$15N budgets to indicate the range in depth and magnitude of productivity within the euphotic zone (Fig. 1c).

Additional changes in response to specific comments are described below.

Review #2 The manuscript by Knapp and colleagues estimates the input of N via N2 fixation using a biogeochemical approach in the western south Pacific Ocean. The manuscript is well-written, and explains the conclusions well, addressing the relevant references. However, this reviewer considers that a few minor changes need to be addressed before publication.

Specific comments. I do agree with Reviewer 1 in two points. First, though it is clear that this manuscript is related to many other publications coming from OUTPACE probably showing a map of station, having the map here will be useful for readers beginning from this work, instead of having to look for the geographical context on their own. Second, it will be useful to extend a bit the context of the sediment traps, maybe adding an additional line in Fig 1 showing either the mixed layer depth or the The units of Average PNsink flux in Table 1 and in the Results section are not the same. The text is in _mol N, and the table in mmol N. I recommend units in agreement in both parts.

As suggested by both reviewers, we have included a map in Fig. 1, as well as additional data, please see the above description. We also thank the Reviewer for their careful attention to detail – the units for the mass flux in Table 1 have been changed so that they are consistent with the units in the text, as well as with the units for N2 fixation, which improves the readability of the manuscript.

Some parts of section 3.2 could be moved to the Methods section, while others seem to fit better in the Discussion, as a first subsection 4.1. My suggestion is that P7 L25 to P8 L14 and P9 L4 to P11 L7 move to the Discussion, while P8 L15 to L23 up to "of the source NO3" move to the Methods. This way the Results subsection is reduced to the description of the results themselves (P8 L23 to P9 L3). P3 L26.

We appreciate the Reviewer's suggestion, and have moved the last two paragraphs of the original Results section to the beginning of the Discussion section as suggested. However, we felt that the paragraph describing NO3-+NO2- $\delta$15N gradients with depth

(in the original manuscript P8 L15 to L23), that the Reviewer suggested be moved to Methods, was better suited to the Results section because it felt out of sequence to move a description of the NO3-+NO2- $\delta$15N gradients with depth to methods before those values were described. Additionally, since we were not interpreting these gradients, just describing the measurements, we felt this text best fit in the Results section. Similarly, we felt that the first paragraph of the Results section was better suited there than in the Discussion, since it describes what the quantitatively relevant terms are in the regional $\delta$15N budgets.

Just curiosity, but is there a reason for using the term Oxygen Deficient Zones instead of the mosth widely used Oxygen Minimun Zones (OMZs)?

Oxygen deficient zones (ODZs) has become the preferred term to identify water columns "where the water column oxygen concentration is so low (low nanomolar range) that oxygen respiration is precluded and denitrification and other low-oxygen (suboxic) metabolisms predominate" (Devol, 2015, Annual Reviews of Marine Science)", and is used to differentiate from other water columns which have higher oxygen concentrations, but all of which have a minimum in oxygen concentrations at some depth that typically coincides with the depth of peak rates of oxic respiration (i.e., remineralization).

P5-P6. The description of the geochemical tools could be moved to the Methods section. And it could be more intuitive to begin the name of the variables by _15N-xx. It is a bit confusing reading NO3+NO2 _15N, for instance.

The text has been moved as suggested.

P7 L16. What do the authors mean with "thermocline NO3+NO2"? Do they refer to subsurface NO3+NO2 as in section 3.2, or NOx produced in the thermocline?

Here we refer to NO3-+NO2- in the thermocline, i.e., between roughly 100 and 800 m, which is plotted in Fig. 1c. This depth range records the isotopic signature of NO3-

reduction processes that occurred in the ODZs of the eastern tropical South Pacific. However, in section 3.2 when we use the term "subsurface", it is to refer to the upper portion of the thermocline, i.e., below the euphotic zone, and specifically the depth range over which the majority of the $NO_3^-+NO_2^-$ that fuels phytoplankton growth in the euphotic zone is sourced.